# Protection of Prisoners with Mental Health Disorders in Italy: Lights and Shadows after the Abolition of Judicial Psychiatric Hospitals

**DOI:** 10.3390/ijerph19169984

**Published:** 2022-08-12

**Authors:** Giulio Di Mizio, Matteo Bolcato, Gianfranco Rivellini, Michele Di Nunzio, Valentina Falvo, Marco Nuti, Francesco Enrichens, Luciano Lucania, Nunzio Di Nunno, Massimo Clerici

**Affiliations:** 1Forensic Medicine and Criminology, Department of Law, Magna Graecia University of Catanzaro, 88100 Catanzaro, Italy; 2Department of Neuroscience, University of Padua, 35121 Padua, Italy; 3Health Unit, REMS of Veneto, ULSS 9 Scaligera—Nogara, 37122 Verona, Italy; 4Center of Mental Health (CSM) “Boccea”, Via di Boccea 271, 00166 Rome, Italy; 5Law and Criminological Firm, 88046 Lamezia Terme, Italy; 6National Agency of Regional Service AGENAS, 30165 Rome, Italy; 7Italian Society of Penitentiary Medicine (SIMSPE), 30165 Rome, Italy; 8Department of History, Society and Studies on Humanity, University of Salento, 83100 Lecce, Italy; 9Psychiatry Unit, School of Medicine and Surgery, University of Milano-Bicocca, 20126 Milan, Italy

**Keywords:** health, prison, prison medicine, REMS, forensic psychiatric treatment, safety, ECHR

## Abstract

In Italy, a person suffering from a mental disorder who commits a crime will be given a custodial security order and serve the period of admission at a *Residenza per la esecuzione delle misure di sicurezza (REMS)* (Residence for the Execution of Security Measures, hereinafter “REMS”). These institutions have been established recently and though equipped with the necessary safety measures, the focus is on psychiatric therapy. Despite being present on a national scale, access is very limited in terms of capacity. Immediate remedial measures are needed, so much so that the European Court of Human Rights recently condemned Italy for this very reason. This article, through a review of the constitutive principles of these institutions, shows how they have very positive aspects such as the attention to necessary psychotherapy in order to protect the right to health and the real taking charge of the fragility of the subjects; however, it is seen how there are many negative aspects linked above all to the scarce availability of places in these structures. The article provides suggestions on a more comprehensive strategy for facilities for detainees with mental disorders.

## 1. Introduction

In Italy, a person responsible for a crime but judged not guilty or guilty but with diminished criminal responsibility by reason of total or partial insanity will be given a custodial security order in the event the person poses “a threat to society”, psychiatrically speaking. To arrive at this assessment, which significantly affects the treatment of the detainee, an accurate medical–legal and psychiatric analysis is necessary, psychiatry specialists will be able to evaluate in this sense whether the pathology is so serious as to constitute a total or partial mental insanity and whether their conduct and capacity for self-control is lost in such a serious way as to constitute a danger to others and to society. Today, such an order means that the person will be admitted to a *Residenza per la esecuzione delle misure di sicurezza* (Residence for the Execution of Security Measures, hereinafter “REMS”) [1], whereas prior to 2015, the order would be executed in a judicial psychiatric hospital (*Ospedale Psichiatrico Giudiziario* (*OPG*), hereinafter “OPG”) [2,3].

The transition from the OPG to the REMS system posed considerable difficulties. Decree-Law No. 211, dated 22 December 2011, ordered the closure of OPGs by 31 March 2013 on the basis of a parliamentary investigation, which confirmed the extreme degradation present in OPGs and the general inefficacy of the treatment interventions that had led to the involuntary commitment of the detainees [4,5].

On 30 July 2008, the Italian Senate set up a select committee [6] to investigate the efficacy and efficiency of the National Health Service, including the quality of life and care within OPGs. The committee’s report on the quality of life and care within OPGs brought to light the deplorable conditions present in these institutions tasked with the custodial care of persons with mental disorders. This report proceeded to describe the “serious and unacceptable structural and hygiene-healthcare deficiencies” in addition to the overall inadequacy of the current system, which is “similar to a prison or mental asylum and completely different from the system used by the Italian mental health service”.

The committee’s report noted serious deficits in the staffing levels of medical and healthcare personnel as well as “clinical practices which are inadequate and, in some cases, harmful to the dignity of the person, as regards mechanical actions and at times the improper use of psychopharmological agents, contrary to their therapeutic purpose and to the regulations for safe use issued by the Italian Medicines Agency (*AIFA*)” [7]. Thus, the committee deemed legislative intervention necessary and urgent in order to take the management of such institutions away from the prison administration. The aim was to effect a “thorough cleansing” of facilities used for sectioning mentally ill offenders, the ultimate objective being the “complete abolition of this method of treating the guilty but mentally ill, i.e., the permanent closure of OPGs”. The committee recommended legislative intervention to “abolish a situation of fact and of law which is, in many aspects, completely incompatible with the precepts of the Constitution”. The senators who served on the select committee even furnished a draft bill to abolish OPGs and replace them with alternative facilities [8], which was in large part approved.

Before it was decided to convene the select committee, the Prime Minister had already issued a decree (1 April 2008), as part of a general reform of the prison healthcare system, setting out a three-stage program to abolish judicial psychiatric hospitals. The first stage consisted of transferring healthcare management to the regional government in which the facility was located and, at the same time, tasking the Mental Health Departments with jurisdiction over the territory in which the OPG is located to draw up an operating plan to discharge those who have completed their custodial security order. The objective of the second stage was to redistribute those who had been committed to the various OPGs throughout the national territory, thus creating treatment bases in order to return detainees to their home environments. In the third stage, the regional governments in Italy were to take charge of all those committed who originated from the territory of said regional government, resulting in region-based psychiatric care provided to mentally ill offenders.

Subsequently, the Minister of Health’s decree of 1 October 2012 (Requirements for Residential Facilities for People Admitted to Judicial Psychiatric Hospitals and to Care and Custody Homes) specified the minimum structural, technological, and organizational requirements for new residential facilities to be able to receive people who have been given a custodial security order of admission to a judicial psychiatric hospital, i.e., a REMS. The sole scope of these internally managed facilities is healthcare, and the facilities fall under the direction of the regional health service, which is under obligation to ensure patient care and proper safety and supervision.

## 2. From OPGs to REMSs

Having described the gradual overhaul of the system to care for those with mental disorders, it is important to mention the relevant developments in legislation and treatment that have led to the progressive abolition of judicial psychiatric hospitals (OPGs) and the establishment of REMSs [9,10].

In the context of prison healthcare, the Prime Minister’s Decree of 1 April 2008 [11] ordered the transfer of healthcare functions, work relationships, financial resources, equipment, and fixed assets from prison administrations to regional governments. This transfer also affected OPGs.

Law No. 9, dated 17 February 2012 [12], enacting Decree-Law No. 211, dated 22 December 2011, specifically Art. 3(3) of the above law, stipulates that beginning with the permanent closure of OPGs, custodial security orders of admission to judicial psychiatric hospitals or in custodial care homes must be carried out exclusively in designated REMS healthcare facilities, with the exception of those who no longer pose a threat to society, who must then be discharged without delay and assisted by the local Mental Health Department.

Only offenders affected by serious mental disorders can be admitted to REMSs, such as psychotic-spectrum disorders, major depression, or serious personality disorders [13], potentially in comorbidity with other disorders. Such disorders must have manifested in the commission of crimes that are commensurate with the symptoms and exhibit an actual need for high-intensity psychiatric treatment.

Art. 3(3) mentioned above detailed a specific allocation of funds to cover the costs incurred in the process of abolishing OPGs, including the recruitment of qualified personnel to deliver treatment–rehabilitation pathways designed to facilitate the recovery and social reintegration of patients transferred from OPGs. This represents an exception to national regulations regarding curtailing personnel expenditure.

At the time OPGs were operational, it was not uncommon to encounter patients who were sectioned with no appointed date for reintegration as a result of multiple extensions of the custodial security order. In that regard, Law No. 81, 9 May 2014 [14], provides that the time period of provisional or final custodial security orders, including admission to a REMS, cannot exceed the custodial period established by law for the crime committed, taking into consideration the maximum sentence available in law.

Ultimately, in addition to the closure of OPGs, the legislator’s objective was to devise and implement a national treatment network for this type of user by consolidating the facilities at the Mental Health Department’s disposal, enabling them to take charge of said offenders and provide treatment–rehabilitation pathways. In fact, personal treatment rehabilitation plans must be created for every user within 45 days of entering a REMS. Committing a person to a REMS to serve the custodial security order, on behalf of the Department of Prison Administration, is based on the principle of territoriality (usual residence/domicile); this aspect is also proof of the desire to maintain the concept of “community psychiatry”, i.e., care given in the place in which the person lives and in which they have developed, to a greater or lesser extent, a social network.

The process of abolishing OPGs, which thus far has been achieved only in part for a number of reasons, which are beyond the remit of this article, has led to the establishment of 31 REMSs throughout Italy [15], resulting in a total of 760 beds (Table 1).

## 3. Positive Aspects of REMS

The creation and implementation of the REMS system has fundamental implications on the management model employed by the Mental Health Department [16,17], including as regards detention, which can be summarized as follows:The active role of users of psychiatric services. The associationism of family members and users, together with the third sector operating within the area of mental health, has been working towards an advocacy approach for some time with a slow but steady increase in representation. The objective of these parties should be to effect continuous improvement in quality that favors best practices that embody the therapeutic-care alliance between users and practitioners. The synergy that exists between these two parties when participating in social health projects is the result of the development of a “culture of the right to the most effective treatment”, which is obtained by means of a shared approach to personalized care (it is no coincidence that these are termed personal treatment rehabilitation plans) [18].Clinical activities focused on areas of vulnerability, risk factors, disabilities, and recovery, widening the scope of the traditional method focused solely on nosographic identification and diagnostic standardization; in other words, a progressive transition from “disease” to the areas of interest cited above, which enable a greater degree of realignment with existential values and legitimate reintegration in the person’s home environment. As a result, interventions targeting treatment and rehabilitation assume greater value, as opposed to those designed solely for reparative purposes, thus resulting in a more comprehensive biopsychosocial approach to psychiatric disorders [19].The enhancement of community care pathways [20]. Psychiatric interventions more acutely focused on territorial services serve to enhance the primary objective of community mental healthcare, to reinforce deinstitutionalization, and to advance a non-hospital centric vision. In this context, extending the health budget to include mentally ill offenders could (1) facilitate the creation of personalized treatment-rehabilitation pathways and (2) “free up” resources for other Mental Health Department activities. In the current historical, financial, and political climate, it cannot be denied that proper organizational analysis aimed at optimizing a community-based approach to psychiatry may well prove to be the springboard for promoting the effective and efficient use of resources required by the Recovery and Resilience Plan (RRP) issued by the European Union to all member nations.

## 4. Negative Aspects of REMS

In recent years, the creation and implementation of the REMS system has been met with considerable criticism, for example, Order No. 131, dated 24 June 2021, issued by the Italian Constitutional Court. In its ruling, the Constitutional Court (tasked with adjudicating the constitutional legitimacy of laws), in considering the difficulties and shortcomings of admissions to REMS facilities, requested the Ministry of Justice and the Ministry of Health to provide an explanatory report on the matter.

The case originated with the Court of Tivoli, Rome, when a preliminary investigations judge raised the question of constitutional legitimacy due to the fact that the provisions would oust the jurisdiction of the Ministry of Justice in executing custodial security orders by admitting offenders to REMS facilities. In that specific case, the judge had made a provisional custodial security order in a REMS facility in the case of a person, investigated for violence towards or threatening of a public official, who was affected by mental disorder and systematic alcohol abuse. Consequently, the accused was deemed a threat to society, and the judge made the custodial security order and provisionally ordered conditional release to a psychiatric facility until such time as transfer to a REMS became possible.

However, the offender systematically refused to be treated and to abide by the obligations set upon him. Nonetheless, despite the insistence of the public prosecutor, the Department of Prison Administration confessed that nothing could be done about the repeated rejection of transfer requests due to the lack of available REMS beds because they are managed exclusively by the regional health service.

The Court ruled that, since the matter did not concern mandatory healthcare treatment but custodial security orders, made on the basis of a two-fold assessment in terms of the commission of a crime and posing a threat to society, the order must come under the Ministry of Justice’s jurisdiction as the prison administration’s supervisory body.

For that reason, the Court ordered the appropriate authorities to prepare a report outlining the number of REMS, the number of patients admitted, the number of patients on the waiting list and the average waiting time, the number of people given an alternative order, such as conditional release, while waiting to be transferred to one of these facilities, and the form of any coordination between the Ministry of Justice and local health authorities.

A subsequent, but by no means less important, judicial ruling on the topic of REMS facilities was given by the European Court of Human Rights, dated 24 January 2022, which granted the appeal of a young psychiatric patient who had been detained for a long time in Rebibbia Prison (Rome) despite the fact that in January 2019, the Supervisory Magistrate had ordered him to be admitted to a REMS facility under a custodial security order. The order had not been carried out due to the chronic lack of available beds.

In March 2020, the patient appealed to the European Court of Human Rights, simultaneously filing a request for interim measures. On 7 April 2020, the European Court of Human Rights indicated an interim measure under Art. 39 of the Regulations, ordering the Italian Government to arrange for the immediate transfer of the appellant to an appropriate facility and ensure he received treatment congruent with his condition. That measure was only carried out on 12 May 2020.

In its 24 January 2022 judgment, the European Court held that subjecting the appellant to the ordinary prison regime, which continued for almost two years despite the opposition of the psychiatrists treating him, prevented him from receiving the necessary healthcare treatment for his psychopathological condition. This constituted a violation of the prohibition of inhuman and degrading treatment and punishment set out in Art. 3 of the European Convention on Human Rights (ECHR).

The Court also held that the detention of the appellant was unlawful, ruling that his being held in seriously degrading conditions in an ordinary penitentiary institution, combined with the failure to provide personalized treatment and to transfer the patient to a REMS facility, constituted a violation of the right to liberty and security of person under Art. 5 of the ECHR.

Furthermore, the Court held that the Italian legal system was in violation of Art. 5 of the ECHR due to the lack of an appropriate provision to ensure fair compensation for the unlawful deprivation of liberty, as well as of Art. 6 of the ECHR due to the national authorities’ failure to execute the trial court’s ruling to release the appellant.

Finally, the Court held that the considerable delay on the part of the Italian government to implement the interim measure issued by the Court in April 2020 was in violation of the right to individual applications under Art. 34 of the ECHR. In fact, the Italian government took more than one month to transfer the appellant to an appropriate facility. The Court also held that “it is incumbent on [every] government to organize its penitentiary system in such a way as to ensure respect for the dignity of detainees, regardless of financial or logistical difficulties”.

Furthermore, the judgment awarded the appellant compensation for non-patrimonial damages due to the violations of Articles 3 and 5 of the ECHR.

The issue that came before the European Court transcends that appellant’s specific case in that the root problem goes to the essence of the Italian legal system, as highlighted numerous times by the prison administration itself: the chronic lack of available REMS beds. According to the latest statistics published by the National Guarantor for the Rights of Persons Detained or Deprived of Liberty [21], as of February 2021, 770 people who received a custodial security order were waiting to be admitted to a REMS facility, 98 of whom were being unlawfully detained in prison facilities, whereas the remaining 672 were free.

## 5. Conclusions

At the time of the reform, i.e., when Decree-Law No. 211, dated 22 December 2011, was enacted by Law No. 9, dated 17 February 2012, the number of REMS facilities and beds required throughout Italy was set using quantitative parameters, which necessitate reevaluation over time to ensure they continue to meet the actual needs. In fact, the number of persons who have received a custodial security order is significantly greater than the number of available REMS beds, posing an immediate turnover problem to facilities that are unable to ensure sufficient rotation of beds in time to take charge of new patients given similar orders by the Judiciary [22,23]. The inability to accommodate the actual demand has led to the phenomenon of the ever-expanding waiting list. Many factors have contributed to this dilemma, not least the elevated number of provisional custodial security orders. As a result, a proportion of mentally ill offenders either stay at home on license under the responsibility of their families with mandatory Mental Health Center monitoring, while others are provisionally put into Therapeutic Communities or remain in custody in Italian prisons *sine titulo*.

The first few years of post-reform experience have highlighted the need for the Mental Health Department to devise an appropriate mental health treatment program and provide training for practitioners in the sector. It is not simply a case of a lack of availability in terms of numbers, but along with the obvious need to expand the number of facilities, it is essential that the complex dynamics (crime, mental disorder, threat to society, etc.) be evaluated and accommodated in a sequence of activities designed to foster involvement, participation, and exchange. Ideally, this approach should encourage participants of REMS treatment–rehabilitation plans to share the thinking, needs, and expectations of their community-creating, in a word, a “pathway” from antisocial to social and, if possible, even pro-social behavior [24].

The human capital element is essential to the success of this approach, i.e., the practitioners operating within the various environments. The specific activities that REMS workers are called upon to perform entail close and prolonged contact with mental and behavioral disorders that render professional continuity problematic due to the risk of burnout and trauma [25]. For this reason, it is essential for practitioners to develop a good awareness of that contact, supported by training and professional development courses and by a shared view of transversality. The collective opinions of the various mental health professionals–psychiatrists, psychologists, nurses [26,27,28,29], social care workers, professional educators, rehabilitation therapists, etc.—can have the advantage, when needed, of providing innovative solutions within a sector that is always at risk of chronicization and automatization.

Ultimately, interventions to expand and sustain REMS activities—through financial, organizational, and professional resources—not only benefit individual patients but also the wider community, enabling the respect of rights and liberties provided for by the Italian Constitution and the European Convention on Human Rights. For that reason, an emergency plan to expand these facilities is now more important than ever, as it is not simply a case of increasing the number and availability of beds but, above all, of creating a capable system of professionals who put their resources to use in these specific activities, remembering that they also need attention, training, and assistance.

## Figures and Tables

**Table 1 ijerph-19-09984-t001:** Distribution of REMS beds in Italy.

Region	Province	No. of Beds
Abruzzo and Molise	Aquila	20
Basilicata	Matera	10
Calabria	Cosenza	20
Campania	Avellino	20
Campania	Caserta	20
Emilia Romagna	Bologna	14
Emilia Romagna	Parma	10
Emilia Romagna	Reggio Emilia	10
Friuli Venezia Giulia	Pordenone	2
Friuli Venezia Giulia	Trieste	2
Friuli Venezia Giulia	Udine	2
Lazio	Frosinone	20
Lazio	Frosinone	11
Lazio	Roma	20
Lazio	Roma	20
Lazio	Roma	20
Lazio	Rieti	15
Liguria	Genova	20
Lombardia and Valle d’Aosta	Mantova	160
Marche	Pesaro Urbino	25
Piemonte	Cuneo	20
Piemonte	Torino	20
Puglia	Barletta Andria Trani	20
Puglia	Brindisi	18
Sardegna	Cagliari	16
Sicilia	Catania	20
Sicilia	Catania	18 (women)
Sicilia	Messina	20
Toscana and Umbria	Firenze	9
Toscana and Umbria	Pisa	30
Trentino Alto Adige	Trento	10
Veneto	Verona	40

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
