# Peer review of "Protection of Prisoners with Mental Health Disorders in Italy: Lights and Shadows after the Abolition of Judicial Psychiatric Hospitals"

_ijerph, 2022, doi:10.3390/ijerph19169984_

Round 1

Reviewer 1 Report

The Authors have conducted an important and timely analysis of the Italian situation after the closure of OPGs and the opening of REMS. In summary, the current model of how the REMS has been regulated has obvious problems. One of the main issues is the reduced capacity of the facilities, causes users with completely different care needs and completely different levels of dangerousness to be found within the same facility; in addition to the endless waiting lists to access the REMS. Another issue that relates more to the organizational aspect is that of security. The law provides for the exclusive presence of social-sanitary-assistance personnel, but there is a growing number of REMS that employ (unarmed) security personnel within the shelter spaces, for the purpose of maintaining order and preventing assaults on staff and other guests. These are just to name a few. What should certainly not be forgotten is the special status of the person admitted to REMS: these are persons deprived of their liberty by virtue of the application of a security measure and therefore particularly vulnerable.

All the aspects highlighted in the present Commentary by the Authors are important and I find them to be an excellent point of discussion, and for that I recommend the publication.

Author Response

We thank the reviewer very much for his comments which have allowed us to improve our work.

Reviewer 2 Report

The article is interesting and generally, it deserves to be published with some revisions that are suggested below:

1)     Would you please describe what the study finding is in the Abstract, to meet the objective of this article is to analyze the positive and negative aspects of the REMS system and to provide suggestions?

2)     Can you state what the study methodology used is, and how to examine the reliability?

3)     Could you offer more of the last 3 years' references, for example, 2022, 2021, and 2019, since your references only have one for 2019 and 2020?

4)     This article is more suitable as a survey report and published in a related magazine.

Author Response

we want to thank the reviewer who allowed us to improve our text with his comments. Here are brief responses to the information we have given.

1) Would you please describe what the study finding is in the Abstract, to meet the objective of this article is to analyze the positive and negative aspects of the REMS system and to provide suggestions?

Thank you, we have taken steps to make the abstract adherent to the suggestions received

3)     Could you offer more of the last 3 years' references, for example, 2022, 2021, and 2019, since your references only have one for 2019 and 2020?

as required we have added some more recent references. Unfortunately, there have been few contributions on this issue in recent years

2)     Can you state what the study methodology used is, and how to examine the reliability?

4)     This article is more suitable as a survey report and published in a related magazine.

thanks for the comment. The article we have written is in the form of a commentary, therefore an article that wants to show the current conditions of this particular topic from a medico-legal and psychotherapist point of view, underlining the need to adopt solutions for the good of the patients and the nation. It is therefore not an article designed to be a systematic review and for this reason we have not expanded the part of materials and methods. We believe it is important that it be published in this magazine because we believe it is important to submit this problem to readers from all over the world on the subject of public health in order to find ideas also at an international level for solutions.

Reviewer 3 Report

This commentary presents the current state of judicial psychiatry in Italy and the challenges that lie ahead. In particular, it discusses in detail the various adverse effects arising from the lack of capacity in the recently established REMS. This document is a valuable reference for the human rights of persons with mental disabilities and will be useful to many readers.   Although no major revisions are necessary, it would have been desirable to add to the opening paragraphs a description of the stage and manner in which psychiatrists are involved in the Italian courts' decision to refer a case to REMS.

Author Response

we want to thank the reviewer for his comments which made it possible to improve the text. We have accepted the suggestion and added some notes in the introduction.

Round 2

Reviewer 2 Report

This is not an academic journal because lack of the part of materials and methods, but may have value for the particular topic from a medico-legal and psychotherapist point of view, in the case of IJERPH journal is not limited the written in the form of a commentary that I will not any comments for this article.

Author Response

We thank the reviewer for his comments and effort to improve our work. We confirm that this is not a systematic review but the journal also provides for the possibility of publishing commentaries on particular topics of interest in the scientific field. In any case, the article contains a research of the literature present on the particular area of REMS although it is not systematic. Thanks for your kind attention